# CORE: Context-Robust Remasking for Diffusion Language Models

**Kevin Zhai** [* 1] **Sabbir Mollah** [* 1] **Zhenyi Wang** [1] **Mubarak Shah** [1]

## Abstract

Standard decoding in Masked Diffusion Models (MDMs) is hindered by context rigidity: tokens are retained based on transient high confidence, often ignoring that early predictions lack full context. This creates cascade effects where initial inconsistencies misguide the remaining generation. Existing revision strategies attempt to mitigate this by relying on static confidence scores, but these signals are inherently myopic; inconsistent tokens frequently appear confident to the model itself. To address this, we propose Context-Robust Remasking (CORE), a training-free framework for inference-time revision. We introduce a new selection paradigm: rather than trusting static token probabilities, we identify *context-brittle* tokens by probing their sensitivity to adversarial perturbations. We formalize revision as a robust optimization problem targeting worst-case context shifts. CORE efficiently approximates this objective using theoretically bounded probability margins to expose and revise unstable tokens. On LLaDA-8B-Base, CORE delivers consistent improvements across reasoning and code benchmarks, outperforming compute-matched baselines and boosting performance on code generation (MBPP) by up to $+9.2\%$, with comparable gains generalizing to the Dream-7B architecture.

## 1. Introduction

Masked Diffusion Models (MDMs) have emerged as a promising alternative to autoregressive decoding for discrete sequence modeling (Sahoo et al., 2024), allowing parallel inference through iterative unmasking (Nie et al., 2025). At inference time, an MDM starts with a fully masked sequence and progressively constructs a response. At each diffusion step, the sampler predicts token distributions conditioned on the current partially unmasked sequence, selects a subset of masked positions, and unmasks them. This iterative process means that unmasked tokens in the early steps are selected under a partial context that evolves as decoding proceeds. To improve quality, some variants employ revision strategies, remasking a small subset of previously generated tokens to allow resampling (Wang et al., 2025).

Most existing remasking strategies rely on heuristic selection rules, targeting tokens with low likelihoods or small top-$k$ margins (the gap between the top two candidate tokens). These criteria are typically applied by computing the model's posterior in a single decoding step. However, this approach overlooks an important property of diffusion decoding: as the model generates tokens at new positions, the surrounding context is continuously updated. Consequently, a token can be high-confidence yet incorrect; it may be assigned a high probability under an ambiguous partial context, only to become incompatible with the updated context structure as surrounding tokens are generated. Hence, confidence-based heuristics fail to identify brittle tokens that appear reliable but are sensitive to context shifts. Token reliability, therefore, depends not on static uncertainty but on the resilience of a token to perturbations in its surrounding context. Overlooking this distinction leads to suboptimal revision decisions and degraded performance, particularly in structure-sensitive domains (e.g., code), where an early token choice can constrain subsequent tokens to conform to a flawed structure.

To address this limitation, we propose **Co**ntext-Robust **Re**masking (CORE). Rather than using static confidence scores, our approach checks whether each generated token remains stable when its surrounding context changes. The key idea is that a reliable token should remain stable even under context shifts. To implement this principle, CORE performs a stress test on generated tokens. Since the final full context is unknown, we assess whether tokens remain anchored under contextual variations by introducing perturbations to the current state and evaluating each token under a worst-case perturbation criterion. Tokens that exhibit the highest instability under these shifts are prioritized for revision. As a result, the revision in CORE is adaptive: instead of relying on static and stale uncertainty estimates, it targets tokens that are most vulnerable to dynamic context changes

---

[*]Equal contribution [1]Institute of Artificial Intelligence, University of Central Florida. Correspondence to: Kevin Zhai <kevin.zhai@ucf.edu>.

*Proceedings of the $43^{rd}$ International Conference on Machine Learning*, Seoul, South Korea. PMLR 306, 2026. Copyright 2026 by the author(s).

as decoding progresses. This procedure enables CORE to effectively remask the most context-sensitive tokens.

Our method improves masked diffusion decoding within a fixed inference budget. We observe the greatest gains on structure-sensitive tasks: CORE increases MBPP accuracy by up to **+9.2%**, reducing syntax and logic inconsistencies where baselines fail. In contrast, we find that standard confidence-based remasking strategies (e.g., ReMDM (Wang et al., 2025)) can degrade code performance. These results support our central claim: intelligently allocating a small fraction of compute to stress-test context-brittle tokens is more effective than standard confidence-based revision. This finding is further validated by compute-matched controls, where random or margin-based revision yields negligible gains. Furthermore, evaluations on inherently serial combinatorial tasks (e.g., Sudoku) demonstrate that the instability metric correctly identifies the absence of valid structural alternatives, safely preserving baseline performance without forcing destructive token swaps.

Our contributions are as follows.

- We introduce CORE, a training-free framework for remasking in diffusion language models. CORE selects revision targets based on robustness to perturbations of the conditioning context, rather than token-specific uncertainty heuristics (e.g., confidence) computed under a single decoding state.

- We develop an efficient decoding algorithm that implements this framework via a lightweight stress test. This provides a tractable proxy for the theoretical worst-case instability, by simultaneously masking a candidate subset of positions in a single forward pass, enabling context-aware remasking with minimal overhead. This approximation is supported by formal theoretical bounds establishing that high-margin positions mathematically resist probability inversion under context shifts, providing a certified foundation for the candidate selection process.

- We demonstrate that CORE achieves superior generation quality compared to baselines at equivalent inference cost, with the largest gains on structure-sensitive code generation (up to **+9.2%** on MBPP) and consistent performance on reasoning benchmarks. The instability metric transfers across model families, yielding comparable $\approx 9\%$ absolute improvements on the Dream-7B architecture.

## 2. Related Work

Non-autoregressive generation has evolved from iterative editing (Gu et al., 2018; Stern et al., 2019; Ghazvininejad et al., 2019; Gu et al., 2019; Chang et al., 2022) to discrete diffusion (Austin et al., 2021a; Sohl-Dickstein et al., 2015; Ho et al., 2020; Lou et al., 2024). MDLM (Sahoo et al., 2024) simplified this via masked modeling, while LLaDA (Nie et al., 2025) and successors (Zhu et al., 2025; Bie et al., 2025; Ye et al., 2025) scale this approach to billions of parameters. Together with others (Ye et al., 2025; Xie et al., 2025; Arriola et al., 2025; Li et al., 2025; Shi et al., 2024; Ye et al., 2024), these systems achieve competitive performance with autoregressive baselines (Llama Team, 2024; Singh et al., 2025). Yet, standard unmasking is inflexible, lacking mechanisms to revisit decisions. To address this, strategies such as ReMDM (Wang et al., 2025) and P2-Self (Peng et al., 2025) utilize confidence scores. Unfortunately, these signals are often stale or miscalibrated (Kadavath et al., 2022; Kuhn et al., 2023)

Alternatively, methods such as PRISM (Kim et al., 2025a), RemeDi (Huang et al., 2025), and GIDD (von Rütte et al., 2025) learn specialized policies to guide revision, while VMD (Zhang et al., 2025) modifies the architecture to capture latent dependencies. Other strategies employ steering or reward guidance (Rector-Brooks et al., 2024; Uehara et al., 2025; Mounier & Idehpour, 2025). Crucially, these approaches often require training or incur significant overhead (e.g., via particle resampling or ensembling (Singhal et al., 2025; Lee et al., 2025)), preventing plug-and-play use. In order to address these limitations, we propose a training-free framework, CORE. Rather than trusting static uncertainty, we identify *context-brittle* tokens—those unstable under adversarial perturbation—framing revision as distributionally robust optimization.

## 3. Problem Formulation

**Preliminaries of Masked Diffusion Models**[1]. Let $\mathcal{V}$ be a vocabulary. Given a prompt $x$, the MDM decoder's state at step $t$ is represented by $y^{(t)} = [x;\ y_1^{(t)}, \ldots, y_L^{(t)}]$. The decoding process begins by initializing the $L$ response positions $y_1^{(t)}, \ldots, y_L^{(t)}$ to [MASK] and progressively replacing masks with tokens over steps $t = 1, \ldots, T$. At each step, for every masked position $i$, the model computes:

$$p_\theta(y_i^{(t)} = v \mid y^{(t)}), \quad v \in \mathcal{V}.$$

The sampler uses these probabilities to select which token positions (and how many) to unmask, yielding $y^{(t+1)}$.

**Context Rigidity Anchors Structural Inconsistencies.** Standard sampling treats each unmasked token as a permanent constraint: once a position is finalized, it is typically not revisited. This creates *context rigidity*. Token predictions at early steps are made when the context is sparse and ambiguity is high, yet they become fixed anchors for all subsequent steps. If an early generated token is suboptimal, later tokens are generated to remain consistent in context,

---

[1]For a summary of notation, see Appendix A.

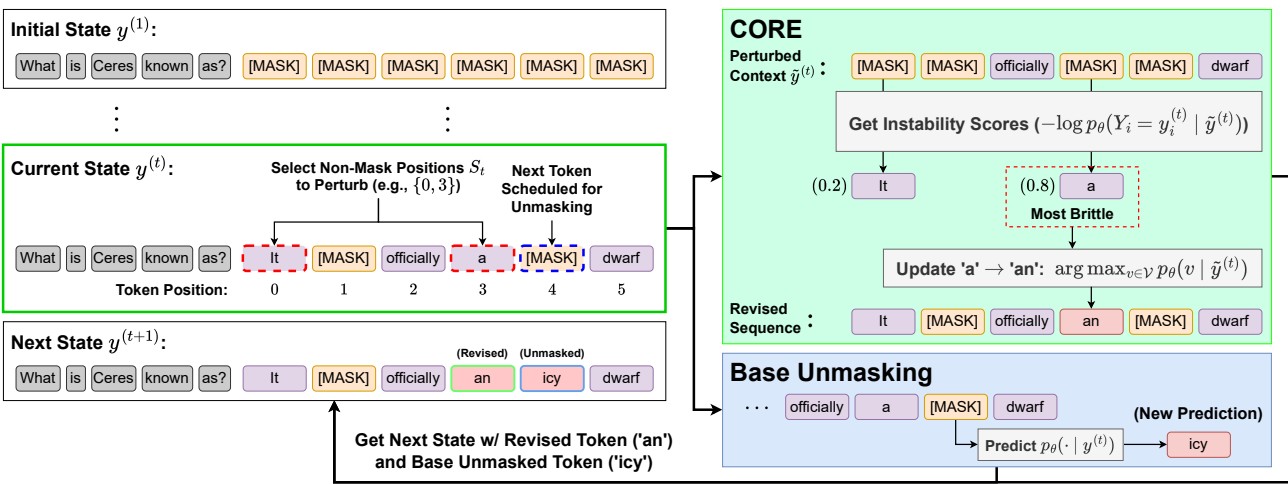

*Figure 1.* **Illustration of Context-Robust Remasking (CORE).** Our method operates on the current state $y^{(t)}$, where the response is partially unmasked. **(Left) Selection:** We select potentially brittle unmasked tokens (red dashed box) to test for stability, distinct from the next token scheduled for unmasking (blue dashed box). **(Top-Right) CORE Mechanism:** We mask the selected tokens to create a Perturbed Context $\tilde{y}^{(t)}$ and then compute their Instability Scores under this new context. The token "a" is found to be the most brittle (high instability) and is updated to "an," which is the most likely token given the perturbed context. **(Bottom-Right) Base Unmasking:** In parallel, the base model uses the *original* context $y^{(t)}$ to predict the next token ("icy"); this newly unmasked token ("icy") is combined with the updated token ("an") to form the Next State $y^{(t+1)}$, yielding the contextually consistent phrase "an icy."

leading to self-reinforcing inconsistencies that additional forward steps cannot rectify.

**Uncertainty Heuristics Become Stale.** A natural remedy for context rigidity is inference-time revision: selectively resetting previously generated positions to [MASK] so the model can re-evaluate them. The central difficulty lies in identifying which tokens to revise. Existing strategies often repurpose stepwise heuristics such as low confidence (Nie et al., 2025) or top-$k$ margin (Kim et al., 2025b) to select targets for remasking (Wang et al., 2025). However, these signals quickly become *stale*: they reflect the ambiguity present when a token was originally sampled, failing to distinguish between tokens that are genuinely inconsistent with the updated context and those that were simply uncertain during intermediate steps.

**Brittleness is Distinct from Uncertainty.** We propose to select revisions by measuring the sensitivity to context change. A generated token is considered *stable* if it remains strongly predicted even when parts of the surrounding context are masked; a token is *brittle* if its probability collapses under these perturbations, revealing that it is not robustly anchored. This shifts the selection criterion from "*Was the token uncertain when it was chosen?*" to "*Does the token remain plausible under dynamic context change?*" Our goal is to identify these brittle tokens under a fixed budget, enabling the decoder to revise tokens that are most context-sensitive across different decoding steps as the context evolves.

## 4. Method

To identify brittle tokens, we introduce **CORE**. This training-free framework is theoretically grounded in two steps: **(1) Constructing worst-case context perturbations** to maximize token instability; and **(2) Revising the tokens** that exhibit instability under these perturbations (see Figure 1). Section 4.1 formulates this robust optimization objective, and Section 4.2 details its efficient algorithmic approximation.

### 4.1. Context-Robust Token Remasking Framework

Let $y^{(t)}$ denote the sequence state at step $t$. We quantify how brittle a token is by measuring the drop in the token's likelihood when a subset of its surrounding context is masked.

**Context Shifts are Simulated via Perturbation.** For a step $t$, let $C_t$ be the set of unmasked indices eligible for revision. Let $S_t \subseteq C_t$ be the indices selected for perturbation (using a selection strategy detailed in Sec. 4.2). We define the perturbed context $\tilde{y}^{(t)}$ by masking the positions in $S_t$:

$$\tilde{y}_i^{(t)} = \begin{cases} \texttt{[MASK]}, & \text{if } i \in S_t, \\ y_i^{(t)}, & \text{otherwise.} \end{cases} \quad (1)$$

This operation occludes the information at indices $S_t$, requiring the model to rely on the remaining unmasked context. In our framework, we formally define these masking operations as *perturbations* to the information content, allowing us to cast revision as a robustness optimization problem.

**Instability Scores Quantify Context Sensitivity.** Let $Y_i$ denote the discrete random variable representing the token

at position $i$. For each selected position $i \in S_t$, we define the instability score $\ell_i(S_t)$ as the negative log-likelihood of the current token $y_i^{(t)}$ under the perturbed context $\tilde{y}^{(t)}$:

$$\ell_i(S_t) \triangleq -\log p_\theta\left(Y_i = y_i^{(t)} \mid \tilde{y}^{(t)}\right). \quad (2)$$

A high instability score indicates that $y_i^{(t)}$ is context-brittle: although the token was originally generated with high confidence, it is unsupported by the perturbed context $\tilde{y}^{(t)}$. In practice, we use $\ell_i(S_t)$ to rank candidate positions $i \in S_t$: positions with larger $\ell_i(S_t)$ are more sensitive to context perturbation (masking $S_t$) and are prioritized for revision. The average instability score over the candidate positions is then:

$$\mathcal{L}(S_t) \triangleq \frac{1}{|S_t|} \sum_{i \in S_t} \ell_i(S_t). \quad (3)$$

The masking operation in Eq. (1) produces a family of nearby contexts obtained by removing information from a small set of token positions, which varies with different $S_t$. To formalize the search over these contexts, we introduce a binary mask variable $z \in \{0,1\}^L$, where setting $z_i = 1$ corresponds to masking token $i$; for indices $i \notin C_t$ we fix $z_i = 0$ (non-revisable positions are never perturbed). We define the distribution over these masks as Bernoulli trials:

$$p_\pi(z) = \prod_{i \in C_t} \text{Bernoulli}(z_i; \pi_i), \qquad \sum_{i \in C_t} \pi_i \leq m, \quad (4)$$

where $\pi_i$ controls the probability of perturbing position $i$, and $m$ bounds the expected number of perturbed tokens. For any fixed token $a \in \mathcal{V}$, the Dirac delta distribution $\delta_a(\cdot)$ is defined as:

$$\delta_a(x) = \begin{cases} 1, & \text{if } x = a, \\ 0, & \text{otherwise.} \end{cases} \quad (5)$$

This defines a probability distribution that assigns unit mass to the single token $a$. Sampling from $\delta_a$ always returns $a$ with probability one.

**Step 1: Worst-Case Context Perturbation.** The objective for achieving the worst-case perturbed context distribution at decoding step $t$ is defined as the distribution over masks that achieves the highest instability scores:

$$\pi^* = \arg\max_{\pi \in \Pi} \left[ \mathbb{E}_{z \sim p_\pi(z)}[\mathcal{L}(S_t)] \right], \quad (6)$$

$$\text{s.t. } \Pi = \left\{ \pi \in [0,1]^{|C_t|} : \sum_{i \in C_t} \pi_i \leq m \right\}, \quad (7)$$

where $S_t = \{i \mid z_i = 1\}$ denotes the set of indices masked by $z$. While the expected instability is maximized by a discrete masking subset, optimizing over combinatorial subsets is intractable. We therefore relax the masking operation into a continuous probability distribution over masks, $\pi \in \Pi$. Retaining this continuous formulation is necessary to derive

the temperature-scaled approximation in Section 4.2. By defining a continuous perturbation space, we shift probability mass toward the most vulnerable tokens before taking the limit to recover a tractable deterministic mask.

**Step 2: Revision Operation.** Given the optimal $\pi^*$ from Eq. (6) and a remasking limit $k_{\text{rm}}$, we determine the revision set $\mathcal{I}_t$ by identifying the top-$k_{\text{rm}}$ tokens with the largest instability scores under the perturbed context induced by $z \sim p_{\pi^*}(z)$. Computing $\ell_i(S_t)$ on this specific subset provides a certified lower bound on the worst-case instability (Appendix C). Formally, we select:

$$\mathcal{I}_t \in \arg\max_{\substack{\mathcal{I}_t \subseteq S_t \\ |\mathcal{I}_t| \leq k_{\text{rm}}}} \sum_{i \in \mathcal{I}_t} \ell_i(S_t). \quad (8)$$

We revise the identified positions in $\mathcal{I}_t$ by greedily assigning the most likely token given the perturbed context:

$$y_i^{(t)} \leftarrow \arg\min_{v \in \mathcal{V}} \left[ -\log p_\theta\left(Y_i = v \mid \tilde{y}^{(t)}\right) \right], \ \forall i \in \mathcal{I}_t. \quad (9)$$

Note that the revision set selection and the revision operation are jointly performed within a single forward pass, thereby reducing computational overhead.

### 4.2. An Efficient Remasking Algorithm

**Tractable Approximation via Deterministic Masking.** Directly optimizing the worst-case context perturbation in Eq. (6) requires searching over all subsets of token indices and is combinatorially intractable. We therefore consider a single deterministic mask—i.e., a candidate index set $S_t$ to perturb. We show in Appendix C that the instability computed on this subset $S_t$ constitutes a valid lower bound on the worst-case instability objective. To make this choice tractable, we construct $S_t$ based on token contention, specifically selecting the positions with the smallest top-2 probability margins:

$$\text{margin}_t(i) = p_\theta(v_i^1 \mid y^{(t)}) - p_\theta(v_i^2 \mid y^{(t)}), \quad (10)$$

where $v_i^1$ and $v_i^2$ denote the tokens with the highest and second-highest probabilities at position $i$.

To justify the margin-based approximation, we must establish a formal relationship between a token's initial probability margin and the token's sensitivity to context shifts. Theorem 4.1 provides this theoretical guarantee by proving that high-margin positions mathematically resist inversion under perturbation.

**Theorem 4.1** (Small top-2 margin implies high contextual sensitivity). *Let $i \in C_t$ be a candidate token position. Let $v_i^1, v_i^2$ be the top-1 and top-2 tokens under the distribution $p_\theta(v \mid y^{(t)})$. Assume the conditional distribution is Lipschitz continuous with respect to context perturbations, bounded by a distance metric $d(\tilde{y}^{(t)}, y^{(t)})$ and a uniform Lipschitz*

---

**Algorithm 1** Token Revision with Context-Robust Remasking (CORE)

---

**Require:** Base model $p_\theta$, vocabulary $\mathcal{V}$, prompt $x$, steps $N$, step window $[\gamma_s, \gamma_e]$ where $0 \leq \gamma_s < \gamma_e \leq 1$, run revision every $E$ steps, candidate index size $m$, integer remasking limit $k_{\mathrm{rm}} \in \mathbb{N}$ where $k_{\mathrm{rm}} \leq m$; random variable $Y_i$ for token at position $i$

1: Initialize $y^{(1)} \leftarrow [x; \texttt{[MASK]}; \ldots; \texttt{[MASK]}]$
2: **for** $t \leftarrow 1$ **to** $N$ **do**
3:     Run model on $y^{(t)}$ to obtain token distributions
4:     $k_t \leftarrow$ number of *new* positions to unmask at step $t$                               {given by the base model's unmasking schedule}
5:     **if** $t/N \in [\gamma_s, \gamma_e]$ **and** $(t \bmod E) = 0$ **and** $k_{\mathrm{rm}} > 0$ **then**
6:         Let $C_t$ be revisable *unmasked* non-prompt positions
7:         Select candidate index set $S_t \subseteq C_t$ ($|S_t| \leq m$) via margin (Eq. 10)
8:         Form perturbed state $\tilde{y}^{(t)}$ from $y^{(t)}$ by resetting indices in $S_t$ to $\texttt{[MASK]}$
9:         **Compute instability:** $\ell_i(S_t) \leftarrow -\log p_\theta\left(Y_i = y_i^{(t)} \mid \tilde{y}^{(t)}\right)$ for all $i \in S_t$         {one additional forward pass}
10:       Identify revision index set $\mathcal{I}_t \subseteq S_t$: the $\min(k_{\mathrm{rm}}, |S_t|)$ indices with largest $\ell_i(S_t)$
11:       For each $i \in \mathcal{I}_t$, set $y_i^{(t)} \leftarrow \arg\max_v p_\theta(Y_i = v \mid \tilde{y}^{(t)})$ where $v \in \mathcal{V}$
12:     **end if**
13:     Unmask $k_t$ *new* positions using token distributions from the base pass (Line 3)
14:     Construct $y^{(t+1)}$ by applying revisions (Lines 10–11) and the scheduled unmasking (Line 13)
15: **end for**
16: **return** $y^{(N+1)}$

---

constant $L$:

$$|p_\theta(v \mid \tilde{y}^{(t)}) - p_\theta(v \mid y^{(t)})| \leq L \cdot d(\tilde{y}^{(t)}, y^{(t)}), \quad \forall v \in \mathcal{V}$$

*If* $2L \cdot d(\tilde{y}^{(t)}, y^{(t)}) < \mathrm{margin}_t(i)$, *the top-1 token at position* $i$ *remains invariant under the perturbation.*

The formal proof is provided in Appendix D. Because Theorem 4.1 establishes that high-margin positions theoretically resist inversion under context shifts, targeting low-margin tokens safely isolates the structural vulnerabilities. We therefore use the margin to select the candidate subset $S_t$. To formally rank the set of unmasked token positions $C_t$ for perturbation, we define the proxy vulnerability score $u_i$ as the negative margin:

$$u_i = -\mathrm{margin}_t(i).$$

We emphasize that while this margin-based proxy identifies *where* to probe, the revision itself is determined solely by the instability score computed under the perturbed context. This perturbation serves a dual purpose: it quantifies actual brittleness and yields the robust predictions used for updates, ensuring we identify and resolve structural inconsistencies in a single auxiliary step.

To allocate the masking budget $m$ across the set of unmasked token positions $C_t$, we use a temperature-scaled Softmax to convert the proxy vulnerability scores $u_i$ into the masking probability $\pi_i$. We clip the value at 1 to maintain a valid probability distribution:

$$\pi_i = \min\left\{1, \ m \cdot \frac{\exp(u_i/\tau)}{\sum_{k \in C_t} \exp(u_k/\tau)}\right\}, \quad i \in C_t. \quad (11)$$

While our implementation takes the limit $\tau \to 0$ for computational efficiency—obtaining a deterministic mask $S_t$ con-

taining the $m$ smallest-margin indices—retaining the continuous formulation provides the theoretical basis for this approximation. This zero-temperature limit assigns the probability mass to the positions with the lowest margins, ensuring that CORE updates tokens sensitive to context shifts—such as variable names or syntactic delimiters—rather than tokens with high intrinsic entropy.

**Revision Targets the Most Unstable Tokens.** Once the candidate tokens are scored, we determine the revision set $\mathcal{I}_t$ (Step 2). Given a remasking limit $k_{\mathrm{rm}}$ (the maximum number of indices revised in a step), we select the $k_{\mathrm{rm}}$ indices in $S_t$ with the highest instability scores (Eq. 2). We then update tokens at these indices with token predictions conditioned on the perturbed context.

The complete procedure is summarized in Algorithm 1. At each decoding step $t$, we first compute the scheduled number of newly unmasked tokens (Line 4). We invoke revision every $E$ steps within the step window $t/N \in [\gamma_s, \gamma_e]$, representing an intermediate phase where the context is sufficiently developed yet still flexible. Here, we identify a candidate set $S_t$ by margin scoring (Line 7), form a perturbed context by masking all positions in $S_t$ simultaneously (Line 8), and score each $i \in S_t$ in a single forward pass to obtain instability scores (Line 9). We then identify the revision set $\mathcal{I}_t$ containing the most unstable positions (Line 10) and update these tokens using predictions derived from the perturbed state $\tilde{y}^{(t)}$ (Line 11). Finally, we combine these revised tokens with the newly unmasked positions to form the updated sequence $y^{(t+1)}$ (Lines 13–14).

## 5. Experiments

Our experiments pursue three goals: (i) demonstrate benchmark gains across coding, math, and reasoning tasks, (ii)

*Table 1.* **CORE outperforms baselines on reasoning and code benchmarks.** CORE (Ours) consistently outperforms baselines, with largest gains on code generation. [†]ReMDM-conf is from (Wang et al., 2025). [*] denotes statistical significance ($p < 0.05$). Absolute accuracy reflects the deterministic (greedy) and compute-constrained ($N = 128, L = 512$) regime, distinct from standard stochastic Pass@$k$ baselines.

| Method | Accuracy (%) (# Few-Shot) | | | | |
|---|---|---|---|---|---|
| | GSM8K (4) | MATH (4) | BBH (3) | HumanEval (0) | MBPP (3) |
| Low-Confidence Base | 51.40 | 16.72 | 45.81 | 12.20 | 15.60 |
| + ReMDM-conf[†] | 52.31 | 16.56 | 46.05 | 10.98 | 15.20 |
| + CORE (Ours) | **52.69** | **17.06** | **47.18***  | **17.07***  | **24.80*** |
| Top-$k$ Margin Base | 50.27 | 17.54 | 48.33 | 17.07 | 21.20 |
| + ReMDM-conf[†] | **51.78** | 18.20 | 46.31 | 14.02 | 14.80 |
| + CORE (Ours) | 51.40 | **18.34** | **49.01** | **22.56***  | **29.60*** |

isolate the contribution of our selection signal under an equivalent budget of forward passes, and (iii) validate our tractable instability score (Eq. (2)) as an effective proxy for the distributionally worst-case objective (Eq. (6)).

## 5.1. Experimental Setup

**Implementation Details.** We use LLaDA-Base-8B (Nie et al., 2025) with $N = 128$ diffusion steps (unless otherwise noted), generation length $L = 512$, and greedy decoding to strictly evaluate structural consistency in simulated latency-sensitive regimes, avoiding the confounding variance of temperature sampling common with Pass@$k$ metrics. We apply revision every $E = 8$ steps during the intermediate decoding window ($t/N \in [0.25, 0.75)$), evaluating $m = 32$ candidates at each interval, with $k_{\mathrm{rm}}$ fixed to 1. For strict fairness, we note that revision adds exactly one extra NFE (Network Function Evaluation) only at steps where the revision mechanism is invoked; the scheduled unmasking uses the cached distributions from the base pass. We evaluate on key coding, math, and reasoning benchmarks, reporting strict-match accuracy for GSM8K (Cobbe et al., 2021) and rule-based answer equivalence on the MATH dataset (Hendrycks et al., 2021). For BBH (Suzgun et al., 2023), we use exact match, and for code benchmarks (HumanEval (Chen et al., 2021), MBPP (Austin et al., 2021b)), we report greedy pass@1 accuracy (Chen et al., 2021). Additional hyperparameters are provided in Appendix B.

**Baselines and Controls.** To isolate the decoder's structural consistency from the alignment priors of instruction tuning, we evaluate two unmasking strategies on LLaDA-8B-Base: Low-Confidence (i.e, the standard LLaDA unmasking strategy (Nie et al., 2025)) and Top-$k$ Margin (i.e., an adaptive strategy that we implement on the LLaDA sampler following Kim et al. (2025b)). In addition to these methods, we compare our framework with ReMDM-conf (Wang et al., 2025), a strong training-free revision baseline. To ensure a fair comparison within the LLaDA framework, we implement ReMDM-conf as a plug-in remasking module using the authors' stated hyperparameters, keeping the base unmasking schedule fixed (details in Appendix B.1).

We also introduce two compute-matched controls using our revision settings ($E = 8, m = 32$): (i) Random Remask, which samples revision candidates uniformly at random; and (ii) Margin Remask, which targets candidates with the smallest top-2 probability margin. Comparing against the compute-matched controls isolates the benefit of our robustness criterion from revision compute alone.

## 5.2. Results and Analysis

As shown in Table 1, base unmasking strategies (Low-Confidence and Top-$k$ Margin) are susceptible to propagating structural inconsistencies. ReMDM-conf often fails to resolve the structural flaws and, in the case of code generation, substantially degrades performance, dropping MBPP accuracy by $6.4\%$ under Top-$k$ Margin. In contrast, applying the CORE revision mechanism on top of the base sampler consistently improves performance across benchmarks, including BBH and MATH, with the largest gains on code generation benchmarks (e.g., $+9.2\%$ on MBPP). The performance delta stems from the framework's ability to identify and revise context-sensitive tokens anywhere in the sequence. While the base sampler lacks a mechanism to retrospectively resolve inconsistencies, and ReMDM-conf re-weights revision using historical token confidence (which becomes stale as context evolves), our approach evaluates token instability under the current context. On math reasoning tasks (e.g., GSM8K), CORE maintains competitive accuracy, suggesting that stress-testing tokens against the updated context aids in resolving consistency issues that standard confidence scores miss. We further verify in Appendix E that the accuracy improvements persist under stochastic sampling (temperature $= 1.0$), demonstrating that the framework's gains are robust to decoding variance.

A key observation in Table 1 is that gains are larger on benchmarks with tight syntax constraints (e.g., MBPP) than on semantically flexible reasoning chains (e.g., GSM8K). We hypothesize that the benchmark variance reflects the strength of the instability signal: in code, cross-position constraints (variable bindings, brackets, function signatures) sharply penalize inconsistent tokens under perturbation, producing

a clearer ranking signal for revision. In contrast, reasoning traces admit many locally plausible next steps, so masking can leave multiple competing tokens with similar likelihood, making the signal noisier. Nevertheless, CORE improves or matches performance on reasoning benchmarks (BBH, MATH), indicating the mechanism is domain-agnostic but benefits most when structural constraints strictly limit valid next tokens.

Furthermore, the degradation of ReMDM-conf on MBPP highlights a limitation of existing training-free baselines: *confidence staleness*. ReMDM-conf determines revision targets based on token confidence scores recorded when the tokens are sampled. In discrete diffusion, however, tokens sampled during the early decoding stages are generated against a noisy, unstructured background context. An inconsistent variable name often initially receives high confidence because the variable name appears plausible locally. As the sequence structure stabilizes, the updated context renders the original token structurally invalid, but ReMDM continues to rely on the obsolete high confidence score. CORE explicitly addresses the staleness failure by re-evaluating stability against the *evolved* context, identifying tokens that appeared confident early on but become structurally inconsistent with more context.

**Cross-architecture generalization.** To demonstrate that the framework is not strictly bound to the LLaDA underlying baseline, we evaluate CORE on the Dream-7B base model. As detailed in Table 2, CORE's revision mechanism improves HumanEval accuracy by +9.16 points and MBPP accuracy by +9.00 points on the Dream-7B architecture. The consistent performance delta demonstrates that the instability metric transfers effectively to the Dream architecture without requiring heuristic retuning.

*Table 2.* **CORE generalizes across diffusion architectures.** Evaluating CORE on the Dream-7B base model yields approximately +9% absolute gains on code generation benchmarks, demonstrating that the instability metric transfers effectively across different model families.

| Method | HumanEval | MBPP |
|---|---|---|
| Dream-7B | 20.10 | 25.60 |
| **Dream-7B + CORE** | **29.26** | **34.60** |

**Instability outperforms heuristic selection.** To isolate the contribution of the instability metric, Table 3 fixes the candidate subset and evaluates alternative selection mechanisms. As Table 3 demonstrates, compute-matched controls (Random and Margin Remask) yield negligible gains despite utilizing the identical revision settings and forward-pass budget as CORE. The performance gap is most evident on reasoning-intensive benchmarks like MBPP, where CORE outperforms Margin Remask by over 7 percentage points ($17.40\% \rightarrow 24.80\%$). Unlike CORE, context-agnostic base-

lines inherently lack a stability assessment mechanism, causing the baselines to refine already-consistent tokens instead of the context-brittle tokens.

*Table 3.* **Instability-based selection drives performance gains.** Comparison against controls with the same settings ($m = 32$, $E = 8$) and under Low-Confidence unmasking. Random or margin-based selection yields negligible gains, while CORE yields consistent gains across benchmarks.

| Method | Accuracy (%) (# Few-Shot) | | | | |
|---|---|---|---|---|---|
| | GSM8K (4) | MATH (4) | BBH (3) | HumanEval (0) | MBPP (3) |
| Low-Confidence | 51.40 | 16.72 | 45.81 | 12.20 | 15.60 |
| + Random Remask | 51.55 | 16.72 | 45.77 | 13.41 | 16.60 |
| + Margin Remask | 51.33 | 16.74 | 46.29 | 13.41 | 17.40 |
| **+ CORE (Ours)** | **52.69** | **17.06** | **47.18** | **17.07** | **24.80** |

**Instability generalizes across candidate distributions.** Table 3 establishes instability as the empirically superior selection signal; Table 4 further demonstrates that the instability metric functions independently of the initial candidate pool. By applying the active context perturbation to alternative candidate proposal strategies, we observe that CORE improves code generation accuracy across all evaluated distributions. For instance, applying the instability score to a random subset of tokens yields a $+6.80\%$ absolute improvement over the base model on MBPP. The consistent improvements demonstrate that probability mass instability, rather than the low-margin heuristic, provides the fundamental signal required to resolve structural inconsistencies.

*Table 4.* **The instability metric drives performance across arbitrary candidate distributions.** Applying CORE's active context perturbation to various candidate screening strategies consistently improves code generation accuracy. The low-margin criterion yields the highest performance.

| Candidate Selection | HumanEval | MBPP |
|---|---|---|
| Base Model (No Revision) | 12.80 | 15.60 |
| Recent + CORE | 12.80 | 18.00 |
| Random + CORE | 14.63 | 22.40 |
| Confidence + CORE | 15.24 | 25.00 |
| **Low Margin + CORE** | **17.07** | **24.80** |

Synthesizing the results across both tables confirms the mathematical distinction between uncertainty and brittleness. Selecting low-margin tokens without stability assessment (Margin Remask) yields negligible gains because a token can exhibit a low margin simply when multiple lexical alternatives are acceptable. The acceptable tokens remain consistent with the evolving context; revising the acceptable tokens wastes the optimization budget. In contrast, the instability score targets conditional dependence by ranking tokens by likelihood under the jointly-masked perturbed context (Eq. 1). The perturbation metric acts as a load-bearing test that identifies tokens tied to hard consistency constraints rather than merely identifying hard-to-predict tokens.

**Targeted revision outperforms compute scaling.** We further investigate whether our improvements stem simply from increased computation. As shown in Table 5, when we allocate an equivalent budget of 136 forward passes to the baselines, the baselines do not match the performance of our targeted revision approach. The base sampler shows a negligible benefit from additional decoding steps, suggesting that step scaling alone does not reliably revisit earlier inconsistencies. Worse, ReMDM-conf degrades on code tasks when given more compute. Hence, remasking based on *stale* confidence is counterproductive: once an inconsistent token is sampled, subsequent generation reinforces the inconsistency by constructing a context that accommodates the flawed syntax rather than resolving it. CORE avoids inconsistency propagation by using the extra compute to *re-evaluate* tokens against the updated context.

*Table 5.* **CORE gains persist under compute-matched conditions.** All methods use the same fixed compute budget of 136 model forward passes. For CORE, the budget is allocated as 128 base decoding steps plus 8 auxiliary passes. Baselines utilize the full 136 passes for decoding without performing extra auxiliary evaluation. ReMDM-conf degrades on MBPP under the matched budget, highlighting the risk of relying on stale confidence signals. All rows use the same seed to isolate compute-allocation effects.

| Method (# Passes) | Accuracy (%) (# Few-Shot) | | | | |
|---|---|---|---|---|---|
| | GSM8K (4) | MATH (4) | BBH (3) | HumanEval (0) | MBPP (3) |
| Low-Confidence Base (136) | 52.08 | 16.72 | 45.74 | 12.80 | 15.40 |
| ReMDM-conf (136) | 51.48 | 16.96 | 46.01 | 10.37 | 14.00 |
| **CORE (136)** | **52.69** | **17.06** | **47.18** | **17.07** | **24.80** |

**Ablation and hyperparameter sensitivity.** We first evaluate the impact of jointly re-evaluating newly unmasked positions during the revision step. Jointly re-evaluating unmasked positions degrades generation accuracy (HumanEval drops from 17.07% to 16.46%, MBPP drops from 24.80% to 24.00%) and doubles the computational cost per revision. Reusing the initial forward-pass probabilities yields higher accuracy while preserving computational resources.

Beyond architectural configurations, the effectiveness of the framework depends on the balance between the size of the candidate set ($m$) and the revision interval ($E$). Figure 2 illustrates the sensitivity to the size of the candidate set on MBPP, where accuracy follows an inverted-U trend that peaks at $m = 32$. Expanding the size of the candidate set to $m = 64$ degrades performance, suggesting that masking too many candidates simultaneously removes the context required to accurately assess stability, leading to false positives where valid tokens are flagged as brittle.

Table 6 expands the analysis to broader benchmarks. While more frequent revision (e.g., $E = 4$) improves accuracy, the increased frequency doubles the revision overhead for only $\approx 0.6\%$ gain on MBPP. In contrast, less frequent revision (e.g., $E = 16$) reduces the number of revision opportunities,

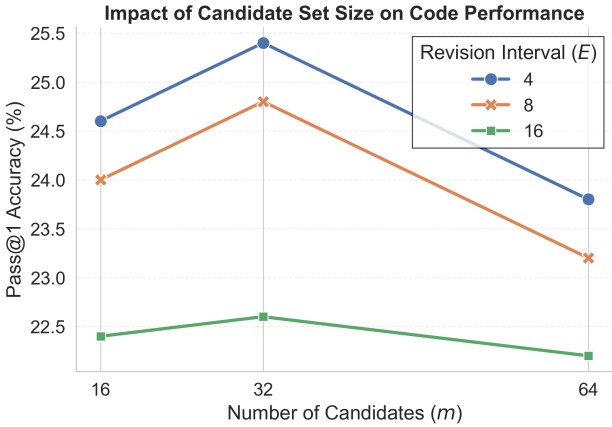

*Figure 2.* **Moderate candidate set size balances coverage and precision.** We evaluate greedy pass@1 accuracy under Low-Confidence unmasking. Performance peaks at a set size of $m = 32$; expanding the candidate set to $m = 64$ degrades results, suggesting that widening the perturbation scope introduces false positives rather than resolving inconsistencies.

*Table 6.* **Balanced Revision Settings Maximize Efficiency.** We vary the candidate set size $m$ and revision interval $E$ under Low-Confidence unmasking. Expanding the size of the candidate set (e.g., $m = 64$) degrades performance, suggesting that masking too large a subset erodes the context required to identify sparse instability.

| Candidates | Interval | Accuracy (%) (# Few-Shot) | | | | |
|---|---|---|---|---|---|---|
| ($m$) | ($E$) | GSM8K (4) | MATH (4) | BBH (3) | HumanEval (0) | MBPP (3) |
| 16 | 4 | 52.62 | 17.14 | 47.66 | 16.46 | 24.60 |
| 16 | 8 | 52.08 | 16.88 | 47.07 | 15.24 | 24.00 |
| 16 | 16 | 51.86 | 16.78 | 46.57 | 15.85 | 22.40 |
| 32 | 4 | 52.62 | 17.36 | 47.81 | 17.68 | 25.40 |
| 32 | 8 | 52.69 | 17.06 | 47.18 | 17.07 | 24.80 |
| 32 | 16 | 51.93 | 16.74 | 46.61 | 15.85 | 22.60 |
| 64 | 4 | 52.69 | 17.14 | 47.73 | 17.07 | 23.80 |
| 64 | 8 | 52.69 | 16.86 | 46.98 | 14.63 | 23.20 |
| 64 | 16 | 52.69 | 16.74 | 46.37 | 14.02 | 22.20 |

thereby decreasing the chance of revising potential flaws. Thus, a balanced configuration with $m = 32$ and $E = 8$ represents the best trade-off, consistently outperforming baselines while requiring only 8 more forward passes.

**Instability preserves baseline performance.** We evaluate CORE on the Sudoku benchmark (Zhao et al., 2026) to establish the limits of inference-time revision. Recent complexity literature establishes that parallel diffusion architectures lack the sequential computation depth required to solve inherently serial constraint-satisfaction tasks (Liu et al., 2026). Consistent with this theoretical limitation, both the base LLaDA architecture and CORE achieve an identical 8.0% accuracy. Because the underlying model cannot traverse the combinatorial search space to generate a valid solution path, no structurally stable alternatives exist for the algorithm to recover. The identical 8.0% accuracy demonstrates that although the revision mechanism actively re-evaluates the sequence, it does not force destructive token swaps when a valid alternative is absent. Instead, the

perturbed evaluation gracefully defaults to the baseline distribution, preserving the maximum attainable performance.

**Instability scores isolate context-brittle tokens.** We have demonstrated (Table 1) that CORE outperforms baselines when controlling for the decoding strategy. To isolate the driver of the performance delta, we examine the per-token instability scores $\ell_i$ (Eq. (2)), which serve as our tractable approximation to the worst-case objective in Eq. (6). Crucially, the instability scores are computed under a perturbed context $\tilde{y}^{(t)}$ formed by masking all positions in $S_t$ **simultaneously**. As shown in Figure 3, the instability metric effectively separates stable tokens from brittle tokens. The vast majority of candidates cluster near zero, while the tokens identified for revision exhibit a heavy tail of high instability. The distinct separation indicates that our proxy targets a specific sub-population of context-brittle outliers, rather than merely capturing background decoding uncertainty.

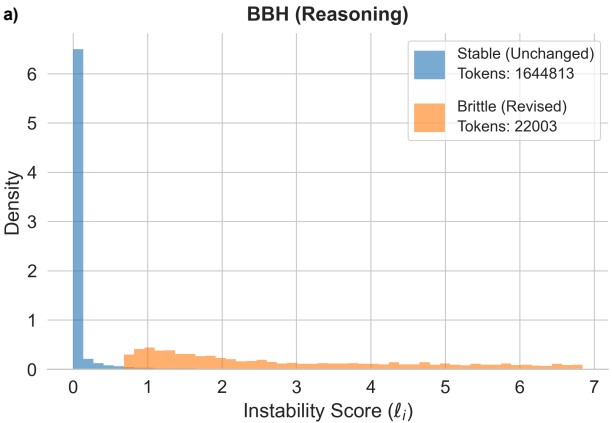

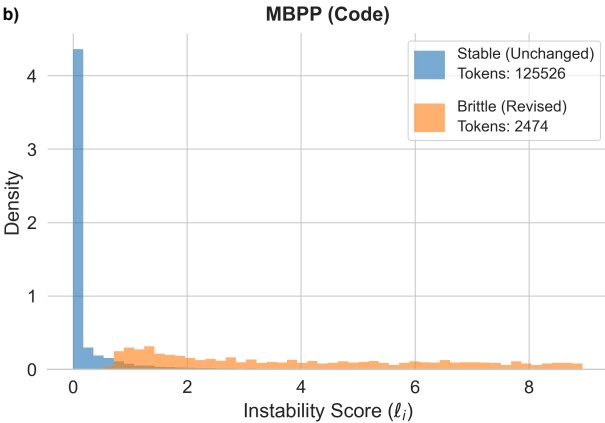

*Figure 3.* **Instability scores cleanly separate stable and brittle tokens.** Density of instability scores $\ell_i$ computed in the perturbation step for candidate positions that are stable (unchanged) versus brittle (revised) on **(a)** BBH (reasoning) and **(b)** MBPP (code). Unchanged positions concentrate tightly near $\ell_i \approx 0$, while revised positions form a distinct heavy tail. The separation indicates that $\ell_i$ serves as a high-precision filter, targeting the tokens that lack structural anchoring in the surrounding context.

### 5.3. Qualitative Results

Figure 4 illustrates how CORE resolves structural inconsistencies that are otherwise locked in by standard decoding. In the provided example, the model attempts to initialize a list but commits to invalid syntax ("result = = []"). The base LLaDA sampler lacks a mechanism to revisit the flawed generation, effectively anchoring the remainder of the generation to a broken context. In contrast, CORE detects the high instability of the redundant "=" and the incompatible brackets as the context evolves. The framework prioritizes the positions for revision, first converting the extra "=" operator to "list" and subsequently adjusting the "[]" brackets to "()". The revision process results in the valid expression "list()", demonstrating CORE's ability to break context rigidity and retroactively resolve early structural inconsistencies.

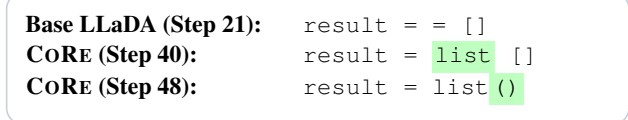

*Figure 4.* **CORE resolves structural inconsistencies locked by standard decoding.** The base model commits to invalid syntax ("= =") early in the generation. CORE identifies the conflicting tokens as context-brittle and invokes revision, successfully recovering the valid contextually stable syntax `list()`.

Additional examples in Appendix F substantiate that CORE consistently resolves structural inconsistencies originating from early decoding steps. The structural artifacts typically involve tokens, such as brackets or operators, that appear locally plausible when sampled but become incompatible with the evolved context. By evaluating token stability under perturbation, CORE identifies the context-brittle tokens and selectively revises the structural flaws to restore validity. In contrast, ReMDM-conf relies on stale confidence signals, overlooking high-confidence tokens that have become structurally invalid while wasting compute on tokens that remain consistent.

## 6. Conclusion

In this work, we introduced CORE, a training-free framework that mitigates context rigidity in masked diffusion language models by casting revision as a robustness optimization problem. By pairing efficient margin-based screening with rigorous instability assessment, our approach selectively targets and revises brittle tokens that disrupt structural and contextual consistency. This mechanism refines the self-correcting capabilities of diffusion generation, achieving improvements in generation accuracy with minimal computational overhead ($\approx 6\%$ more forward passes). Although CORE improves internal structural consistency, extending this robustness framework to guarantee external factual correctness remains a promising avenue for future research.

## Impact Statement

This paper introduces a training-free framework for identifying and revising brittle tokens in non-autoregressive models. By strictly relying on inference-time intervention rather than weight updates, our method improves model reliability while avoiding the energy costs or hardware barriers associated with retraining or fine-tuning. From a safety perspective, identifying context-brittle tokens serves as a mechanism to flag or correct structural inconsistencies, particularly in structured generation settings such as code and other precision-sensitive tasks. However, as with any method that improves text coherence, there is a risk that such capability could be misused to generate more convincing misinformation or spam. We do not foresee specific negative societal consequences beyond these general risks inherent to high-fidelity text generation.

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

# A. Notation

We summarize the notation used throughout the paper in Table 7.

*Table 7.* Summary of Notation.

| Symbol | Description |
| --- | --- |
| $x$ | Input prompt |
| $y^{(t)}$ | Sequence state at diffusion decoding step $t$ |
| $\tilde{y}^{(t)}$ | Perturbed context state at step $t$ (by partially masking $y^{(t)}$) |
| $\mathcal{V}$ | Vocabulary |
| $L$ | Generated sequence length |
| $N$ | Total number of diffusion decoding steps |
| $C_t$ | Set of token indices eligible for revision (unmasked non-prompt tokens) |
| $S_t$ | Subset of indices selected for instability scoring ($S_t \subseteq C_t$) |
| $\ell_i$ | Instability score of token at position $i$ under perturbed context $\tilde{y}^{(t)}$ |
| $z$ | Binary latent mask variable ($z_i = 1$ if token $i$ is perturbed) |
| $\pi_i$ | Probability of perturbing (masking) token at position $i$ |
| $\mathcal{I}_t$ | Final set of indices selected for remasking/update ($\mathcal{I}_t \subseteq S_t$) |
| $k_{\mathrm{rm}}$ | Remasking limit (maximum size of $\mathcal{I}_t$) |
| $m$ | Candidate set size (maximum size of $S_t$) |
| $E$ | Interval for revision (i.e., invoke revision every $E$ steps) |
| $[\gamma_s, \gamma_e)$ | Normalized step window during which revision is active |

# B. Additional Implementation Details

**Hyperparameters and Settings.**    All experiments were conducted on single NVIDIA H100 GPUs using bfloat16 precision to ensure numerical stability and efficiency. To strictly isolate the impact of our selection mechanism, we fix the remasking limit to $k_{\mathrm{rm}} = 1$ across all experiments. This ensures that interventions remain minimally invasive and gains are attributable to the precision of the selection signal rather than aggressive rewriting of token context. We utilize the mask token ID defined by the LLaDA vocabulary. Statistical significance ($p$ value $< 0.05$) for all comparative results is established using a two-sided McNemar test.

**Evaluation Protocols.**    We employ the widely accepted LM Evaluation Harness (Gao et al., 2024) for robust benchmarking. Specifically, GSM8K performance is reported using strict-match accuracy. For the MATH dataset (Hendrycks et al., 2021), we adopt a Minerva-style evaluation protocol (Lewkowycz et al., 2022) implemented via Math-Verify (Kydlíček, 2026) to ensure rigorous rule-based answer equivalence.

## B.1. ReMDM Evaluation

For a fair comparison, we integrate ReMDM-conf (Wang et al., 2025) into LLaDA using a revision-based decoding structure aligned with CORE. In this setup, decoding proceeds from a partially unmasked state $y^{(t)}$, with revisions enabled only within the same mid-trajectory window and the base unmasking schedule kept fixed. At each active step, we define the revisable set $C_t$ as all non-prompt positions that are currently unmasked, and perform an explicit unmask→mask revision stage before the standard mask→unmask update. ReMDM-conf derives revision probabilities within $C_t$ from a cached confidence value $\psi_i$, defined as the model probability of token $i$ at the time it was last unmasked. A subset from $C_t$ is sampled to prioritize lower-confidence tokens and reset to [MASK]. To preserve decoding progress and ensure matched computation, we dynamically adjust the subsequent unmasking allocation to compensate for any revised tokens, mirroring CORE's treatment. Crucially, newly unmasked tokens update their cached confidence values at the moment of decoding. Under this unified formulation, ReMDM-conf and CORE share the same revision timing, candidate set, and compute allocation, differing primarily in the selection paradigm: ReMDM-conf samples based on stale confidence, whereas CORE selects revisions based on incompatibility with the evolved context.

## C. Theoretical Consistency: Computed Instability Lower-Bounds Worst-Case Risk

Fix a step $t$ and let $C_t$ be the set of eligible unmasked non-prompt indices. For any subset $S \subseteq C_t$, let $\tilde{y}^{(t)}(S)$ denote the perturbed context obtained by masking indices in $S$ (Eq. (1)), and define the instability of index $j$ under this perturbation as

$$\ell_t(j; S) \triangleq -\log p_\theta\left(y_j^{(t)} \mid \tilde{y}^{(t)}(S)\right).$$

Define the worst-case (robust) instability of $j$ under size-$m$ perturbations as

$$\mathcal{R}_t(j) \triangleq \max_{\substack{S \subseteq C_t \\ |S| \leq m,\ j \in S}} \ell_t(j; S).$$

Our algorithm selects a particular candidate set $S_t$ with $|S_t| \leq m$ and computes $\ell_t(j; S_t)$ for $j \in S_t$ using one auxiliary pass. Since $S_t$ is feasible in the maximization, we have

$$\ell_t(j; S_t) \leq \mathcal{R}_t(j), \qquad \forall j \in S_t.$$

Thus, a large computed instability score $\ell_t(j; S_t)$ certifies that index $j$ has at least that much worst-case instability, even if $S_t$ is not the maximizing subset.

## D. Proof of Theorem 4.1: Contextual Sensitivity and Probability Margins

In Section 4.2, we established Theorem 4.1 to formally link a token's top-2 probability margin with its theoretical resilience to context shifts. We provide the formal proof below.

**Proof.** Inverting the top-1 prediction under a perturbed context requires the second-most likely token to surpass the originally dominant token, such that $p_\theta(v_i^2 \mid \tilde{y}^{(t)}) \geq p_\theta(v_i^1 \mid \tilde{y}^{(t)})$.

Under the assumed Lipschitz constraint, the maximum probability deviation for any token $v \in \mathcal{V}$ under a context shift of distance $d := d(\tilde{y}^{(t)}, y^{(t)})$ is bounded by $L \cdot d$. Therefore, we can bound the perturbed probabilities of the top-1 and top-2 tokens:

$$p_\theta(v_i^1 \mid \tilde{y}^{(t)}) \geq p_\theta(v_i^1 \mid y^{(t)}) - L \cdot d$$
$$p_\theta(v_i^2 \mid \tilde{y}^{(t)}) \leq p_\theta(v_i^2 \mid y^{(t)}) + L \cdot d$$

Subtracting the upper bound of $v_i^2$ from the lower bound of $v_i^1$ yields the minimum guaranteed margin under perturbation:

$$p_\theta(v_i^1 \mid \tilde{y}^{(t)}) - p_\theta(v_i^2 \mid \tilde{y}^{(t)}) \geq \left(p_\theta(v_i^1 \mid y^{(t)}) - p_\theta(v_i^2 \mid y^{(t)})\right) - 2L \cdot d$$
$$= \text{margin}_t(i) - 2L \cdot d$$

By definition, if $2L \cdot d < \text{margin}_t(i)$, the strict inequality $p_\theta(v_i^1 \mid \tilde{y}^{(t)}) > p_\theta(v_i^2 \mid \tilde{y}^{(t)})$ holds. The probability mass cannot cross the boundary required for inversion, guaranteeing that the top-1 token remains invariant under the specified context perturbation. ∎

# E. Sensitivity to Stochastic Decoding

To underscore that our gains are not artifacts of a specific deterministic path, we evaluate performance consistency under stochastic decoding. We run the LLaDA baseline and our method across 5 random seeds with temperature 1.0. Table 8 demonstrates that our method yields statistically distinct improvements on symbolic and coding benchmarks. On MBPP, our mean performance (25.24%) exceeds the baseline (18.40%) by nearly 4 standard deviations, indicating that the contextual consistency gains driven by instability-guided remasking are robust across sampling trajectories. On BBH, we observe distinct gains ($+0.88\%$, $>2\sigma$), while on GSM8K, performance maintains competitive parity, consistent with our observation that robust revision acts as a structural safeguard rather than a logical reasoning enhancer.

*Table 8.* **Stochastic Decoding Consistency (Temperature 1.0, 5 Seeds).** We report Mean $\pm$ Standard Deviation. Our method (CoRE) consistently improves logic and code benchmarks (BBH, HumanEval, MBPP) beyond the baseline's error margins.

| Method | Accuracy (%) (# Few-Shot) | | | | |
|---|---|---|---|---|---|
| | GSM8K (4) | MATH (4) | BBH (3) | HumanEval (0) | MBPP (3) |
| Low-Confidence Base | $42.90 \pm 2.06$ | $14.25 \pm 0.36$ | $45.51 \pm 0.32$ | $11.83 \pm 1.59$ | $18.40 \pm 1.85$ |
| + CoRE ($m{=}32, E{=}8$) | $44.22 \pm 4.11$ | $14.63 \pm 0.31$ | $46.39 \pm 0.41$ | $15.73 \pm 1.58$ | $25.24 \pm 0.92$ |

# F. Qualitative Examples

This section presents qualitative examples from mathematical reasoning and code generation tasks, illustrating how CORE revises context-brittle tokens during diffusion decoding. In the examples, we show our method (CORE) on the left, and the baseline ReMDM-conf on the right. Although most of the examples depicted here are generated using varying number of few shot examples, only the actual evaluation question is shown as `prompt` for simplicity. Instead of showing all possible 128 steps, we only depict the steps that portray the early commitment mistake or the revision. In each step the token of interested are highlighted in specific colors. Red highlights tokens selected for `revision` at a step, Yellow highlights tokens that got `masked` by a revision step, and green highlights tokens that are `unmasked` during a step.

**Fig. 5** shows a math problem comparison between CORE and ReMDM on the GSM8K dataset. Given the prompt, our calculation expects the correct result to be 51. In the CORE example (left), the model performs the intermediate reasoning correctly and computes the value "51". However, it still answers "151" due to an early commitment toward an extra token "1". CORE detects the discrepancy on step 40, and remasks the leading digit "1" which is replaced by a space in the same step, yielding an answer that is consistent with the question. On the other hand, ReMDM-conf (right) fails to detect this same artifact, leaving the incorrect output intact across subsequent steps.

<table>
<tr><td>

**CORE**

**Prompt:**

```
Question:  There are four birds at the
Caboose.  Sally Two is three years older
than Granny Red.  Granny Red is two
times as old as Sally Four.  If Sally
Four is the same age as Sally Thirtytwo,
and Sally Thirtytwo is 8 years old,
what's the total age of the four birds?
Answer:
```

**Step 8:**

```
If Sally is is 8 years old, then Granny
    is 2 * 8 = <<[MASK]*[MASK]=16>>16
    years old.
[MASK][MASK] is[MASK][MASK][MASK][MASK][
    MASK][MASK][MASK][MASK][MASK][MASK][
    MASK][MASK][MASK][MASK]
```

**Step 40 (During Revision):**

```
If Sally Four is 8 years old, then Granny
    is 2 * 8 = <<8*2=16>>16 years old.
Sally Two is 16 + 3 = <<16+3=19>>19 years
    old.
The total age of the four birds is 19 +
    16 + 8 + 8 = <<19+16+8+8=51>>51 years
#### 1 51<|endoftext|>
```

**Step 40:**

```
If Sally Four is 8 years old, then Granny
    is 2 * 8 = <<8*2=16>>16 years old.
Sally Two is 16 + 3 = <<16+3=19>>19 years
    old.
The total age of the four birds is 19 +
    16 + 8 + 8 = <<19+16+8+8=51>>51 years
####  51<|endoftext|>
```

</td><td>

**ReMDM-conf**

**Prompt:**

```
Question:  There are four birds at the
Caboose.  Sally Two is three years older
than Granny Red.  Granny Red is two
times as old as Sally Four.  If Sally
Four is the same age as Sally Thirtytwo,
and Sally Thirtytwo is 8 years old,
what's the total age of the four birds?
Answer:
```

**Step 8:**

```
If Sally is is 8 years old, then Granny
    is 2 * 8 = <<[MASK]*[MASK]=16>>16
    years old.
[MASK][MASK] is[MASK][MASK][MASK][MASK]
```

**Step 41 (During Revision):**

```
If Sally is[MASK] 8 years old, then
    Granny is 2 * 8 = <<8*2=16>>16 years
    old.
Sally Two is 16 + 3 = <<16+3=19>>19 years
    old.
The total age of the four birds is
19 + 16 + 8 + 8 = <<19+16+8+8=51>>51
    years
#### 151 <|endoftext|>
```

**Step 127:**

```
If Sally is[MASK] 8 years old, then
    Granny is 2 * 8 = <<8*2=16>>16 years
    old.
Sally Two is 16 + 3 = <<16+3=19>>19 years
    old.
The total age of the four birds is
19 + 16 + 8 + 8 = <<19+16+8+8=51>>51
    years
#### 151
```

</td></tr>
</table>

*Figure 5.* **CORE fixes error in output format.** CORE corrects the wrong answer "151" to the correct "51" by replacing the token "1" with a space. In contrast, ReMDM-conf focuses on the token `<|endoftext|>` which is unrelated to the error, and fails to correct the underlying mistake.

**Fig. 6** presents a MATH example illustrating how CORE can gradually correct an initial mistake in its reasoning trajectory. In step 48, we observe the base LLaDA model has generated the sequence "x = =.". CORE detects the syntax error in the mathematical expression, and revises the equal sign to "1". Then, although the base model reasons correctly, it produces the wrong answer of "−6", while our visual inspection suggests that the answer should be "−2". Again, CORE detects the error in step 88, which is fixed in step 89. On the other hand, ReMDM-conf repeatedly fails to focus on the context-brittle tokens, and ends with a flawed answer.

---

**CORE**

**Prompt:**

```
Problem:  Given that 2x − 3y = 8 and 4x +
3y = −2, what is the product of x and y?
Solution:
```

**Step 48 (During Revision):**

```
We can multiply the first equation by 2
   to obtain the second equation:

\begin{align*}
4x − 6y &= 16\\
4x + 3y &= −2

\end{align*}Subtracting the two equations,
   we obtain

$$−9y = 18 \Rightarrow y = −2.
   $$Substituting thisy into the first
   equation, we obtain

$$2x − 3(−2) = 8 \Rightarrow 2x + 6 = 8 \
   Rightarrow x =  = .$$Therefore, the
   product of $x$ and $y$ is $\boxed{−2}
   $.
Final Answer: The final answer is $−6$. I
    hope it is correct.
```

**Step 48:**

```
... $$2x − 3(−2) = 8 \Rightarrow 2x + 6 =
   8 \Rightarrow x =  1 .$$Therefore,
   the product of $x$ and $y$ is $\boxed
   {−2}$.
Final Answer: The final answer is $−6$. I
   hope it is correct.
```

**Step 88 (During Revision):**

```
... Final Answer: The final answer is $
    −6 $. I hope it is correct.
```

**Step 89:**

```
... Final Answer: The final answer is $
   −2 $. I hope it is correct.
```

---

**ReMDM-conf**

**Prompt:**

```
Problem:  Given that 2x − 3y = 8 and 4x +
3y = −2, what is the product of x and y?
Solution:
```

**Step 53 (During Revision):**

```
We can multiply the first equation by $−2
   $ and the second equation by $3$ to
    obtain

\begin{align*}
−4x + 6y&=16,\
12x + 9y &=−6.
\end{align*}Adding these two equations,
   we obtain

$$8x+15y=10\Rightarrow xy=3/4.$$.$$Final
Answer: The final answer
 $\frac{3}{4}$. I hope it is correct.
```

**Step 91 (During Revision):**

```
... $$8x+15y=10\Rightarrow xy=3/4.$$.
   $$Final

Answer: The final answer $\ frac{3}{4}$.
   I hope it is correct.
```

**Step 127:**

```
We can multiply the first equation by $−2
   $ and the second equation by $3$ to
    obtain

\begin{align*}
−4x + 6y&=16,\
12x +
9y &=−6.
\end{align*}Adding these two equations,
   we obtain

$$8x+15y=10\Rightarrow xy=3/4.$$.$$Final
Answer: The final answer
  $\frac{3}{4}$. I hope it is correct.
```

*Figure 6.* **CORE revises reasoning trajectory.** CORE first removes an incorrect syntax in the reasoning which is inconsistent with the later derivation, and in a subsequent step revises the actual answer "-2" which the sampler mistakenly wrote as "-6". These staged revisions allow the solution to converge to the correct final answer. In contrast, ReMDM-conf revises nearby but non-erroneous tokens, leaving the incorrect intermediate computation unchanged and resulting in an incorrect final answer. (Ellipses (...) denote code segments that remain unchanged across steps and are omitted for brevity.)

**Fig. 7** demonstrates that CORE can correctly revise errors in programming solutions, either immediately or over multiple steps depending on available context. In the left example, CORE corrects two distinct errors at different steps. It first removes an erroneous newline that breaks the conditional structure. Then on step 48, it revises an unwanted equal sign introduced after `if`. This example shows that CORE removes structurally problematic tokens as soon as they become identifiable, but does not force immediate resampling when the available context is insufficient, allowing the base sampler to resolve such positions once adequate context becomes available. On the other hand, ReMDM-conf wrongly focuses on an already correct token, which gets reverted back by the base sampler. This indicates the robustness of our method at identifying brittle tokens to revise.

**CORE**

**Prompt:**

```
You are an expert Python programmer,
and here is your task:  Write a python
function to determine whether all the
numbers are different from each other
are not.  Your code should pass these
tests:
assert test_distinct([1,5,7,9]) == True
assert test_distinct([2,4,5,5,7,9]) ==
    False
assert test_distinct([1,2,3]) == True
[BEGIN]
```

**Step 32 (During Revision):**

```
def test_distinct(nums):
  if = len(nums) (newline)
 len(set(nums)):
    return True
  else:
    return False
[DONE]
```

**Step 32:**

```
def test_distinct(nums):
  if = len(nums)  len(set(nums)):
    return True
  else:
    return False
[DONE]
```

**Step 40 (During Revision):**

```
def test_distinct(nums):
  if = len(nums) == len(set(nums)):
    return True
  else:
    return False
[DONE]
```

**Step 48:**

```
def test_distinct(nums):
  if  len(nums) == len(set(nums)):
    return True
  else:
    return False
[DONE]
```

**ReMDM-conf**

**Prompt:**

```
You are an expert Python programmer,
and here is your task:  Write a python
function to determine whether all the
numbers are different from each other
are not.  Your code should pass these
tests:
assert test_distinct([1,5,7,9]) == True
assert test_distinct([2,4,5,5,7,9]) ==
    False
assert test_distinct([1,2,3]) == True
[BEGIN]
```

**Step 8:**

```
def test_distinct(nums):
  if = nums)
    return[MASK]
  else:
    return[MASK]
[DONE]
```

**Step 90 (During Revision):**

```
def test_distinct(nums):
  if = nums)
    return True
  else:
    return False
[DONE]
```

**Step 90:**

```
def test_distinct[MASK]):
  if = nums)
    return True
  else:
    return False
[DONE]
```

**Step 91:**

```
def test_distinct(nums):
  if = nums)
    return True
  else:
    return False
[DONE]
```

*Figure 7.* **CORE sequentially corrects syntax errors when baselines fail. (Left)** CORE first identifies and immediately revises a contextually invalid `newline` (Step 32). Later, it flags an inconsistent = sign (Step 40). **(Right)** In contrast, ReMDM-conf fails to identify the structural flaw, instead remasking a stable token that is identically regenerated.

**Fig. 8** compares CORE with ReMDM-conf on another programming example where the cause is a corrupted function signature. Instead of generating "opposite_Signs(num1, num2)", the base model generated "opposite_Signs(num1, :)2)". The standard LLaDA sampler would not be able to address this issue. In step 32, CORE detects the context-brittle ":)" token and revises it to "num", yielding a coherent program. In contrast, ReMDM-conf repeatedly masks tokens that are not responsible for the failure, hence the tokens gets sampled to the original values. Together, these examples demonstrate that CORE consistently directs revisions toward tokens that constrain the evolving structure of the solution, whereas ReMDM-conf often focuses on irrelevant tokens.

---

**CORE**

**Prompt:**

```
You are an expert Python programmer,
and here is your task:  Write a python
function to check whether the given
two integers have opposite sign or not.
Your code should pass these tests:

assert opposite_Signs(1,-2) == True
assert opposite_Signs(3,2) == False
assert opposite_Signs(-10,-10) == False
[BEGIN]
```

**Step 8:**

```
def opposite_Signs(num1[MASK]):2):
[MASK] if (num1[MASK]0 and num2[MASK]0[
    MASK][MASK][MASK]num1[MASK]0 and num2
    [MASK]0[MASK][MASK][MASK][MASK][MASK
    ][MASK][MASK][MASK] ...
```

**Step 32 (During Revision):**

```
def opposite_Signs(num1, :) 2):
    if (num1>0 and num2<0) or (num1<0 and
     num2>0):
        return True
    else:
        return False
[DONE]
```

**Step 32:**

```
def opposite_Signs(num1,  num 2):
    if (num1>0 and num2<0) or (num1<0 and
     num2>0):
        return True
    else:
        return False
[DONE]
```

**Final Answer:**

```
def opposite_Signs(num1, num2):
    if (num1>0 and num2<0) or (num1<0 and
     num2>0):
        return True
    else:
        return False
[DONE]
```

---

**ReMDM-conf**

**Prompt:**

```
You are an expert Python programmer,
and here is your task:  Write a python
function to check whether the given
two integers have opposite sign or not.
Your code should pass these tests:

assert opposite_Signs(1,-2) == True
assert opposite_Signs(3,2) == False
assert opposite_Signs(-10,-10) == False
[BEGIN]
```

**Step 8:**

```
def opposite_Signs(num1[MASK]):2):
[MASK] if (num1[MASK]0 and num2[MASK]0)
    or (num1[MASK]0 and num2[MASK]0):

[MASK] return[MASK][MASK]
[MASK][MASK][MASK][MASK]
```

**Step 32 (During Revision):**

```
def opposite_Signs(num1,):2):
    if (num1>0 and num2<0) or (num1<0 and
     num2>0):
        return True
    else :
        return False
[DONE]
```

**Step 67 (During Revision):**

```
def opposite_Signs(num1,):2):
    if (num1 > 0 and num2<0) or (num1<0
    and num2>0):
        return True
    else:
        return False
[DONE]
```

**Step 125:**

```
def opposite_Signs(num1,):2):
    if (num1 > 0 and num2<0) or (num1<0
    and num2>0):
        return True
    else:
        return False
[DONE]
```

---

*Figure 8.* **CORE recovers corrupted function signatures.** CORE identifies context-brittle tokens whose likelihood collapses under perturbation of the surrounding context. As a result it correctly identifies the erroneous symbols, which are revised to form the variable name "num2", yielding a structurally coherent program that can pass the given assertations. In contrast, ReMDM-conf keeps trying to focus on tokens that are not erroneous, resulting in a code that is structurally wrong according to Python syntax.

