# OpenReview forum: "CORE: Context-Robust Remasking for Diffusion Language Models"
_ICML.cc/2026/Conference — ICML 2026 regular_

### Official Review · Reviewer_HGzu · 2026-03-08

**Soundness:** 3
**Presentation:** 3
**Significance:** 3
**Originality:** 3
**Overall Recommendation:** 5
**Confidence:** 2

**Summary:**

This paper proposes CoRe, a remasking mechanism for diffusion language models. Instead of choosing the remasked positions based on confidence, as in existing works, the authors proposed to choose remasked positions based on context-stability -- positions whose output distribution changes a lot when the context changes. The authors further proposed a fast approach to choosing the perturbed context. Experiments show that the proposed remasking significantly improves diffusion language models' performance on coding and reasoning tasks. Further ablation studies confirm that the improvement is due to the improved metric for selecting remasking positions.

**Compliance With Llm Reviewing Policy:**

Affirmed.

**Final Justification:**

The rebuttal has addressed the writing concerns.

**Key Questions For Authors:**

See weakness

**Limitations:**

Yes

**Strengths And Weaknesses:**

**Strengths:**

1. The proposed method shows a very great improvement in math and coding tasks over baselines.
2. Sufficient ablation studies are performed

**Weaknesses**

My biggest concern with the paper is that the math formulations can be improved. Specifically

1. On page 2, the equation
$$
p_{\theta} (y_i^{(t)} = v | y^{(t)}, i)
$$
does not look correct, because it involves the distribution of a variable conditional on itself (the authors used the same variable notation for both the conditioning and the distribution). I understand what the authors intend to show with this notation, but this can be expressed in a more rigorous way. For example, the author could use the unmasked subset of $y^{(t)}$ as the condition, or could use a different variable to distinguish the unmasked or masked version of the variable.

2. Equation (11), $p_{\theta}(v|\tilde{y}^{(t)})$ seemed very ambiguous because it does not specify the position $i$. I would recommend changing it back to $p_{\theta}(y_i^{(t)} = v|\tilde{y}^{(t)})$.

3. I wonder why Equation (8) is expressed in such a circuitous way. Considering the optimal solution for $\pi$ is a corner solution concentrated on the set $S_t$ this maximizes $\mathcal{L}(S_t)$. A more straightforward formulation seems to be
$$
{S_t}^* = \text{argmax}_{S_t \subset C_t} \mathcal{L}(S_t), s.t. |S_t| \leq m
$$
This formulation maintains consistency with Equation (10). Is there any reason to have disparate formulations for Equations (8) and (10)?

4. Perhaps one reason the authors introduced $\pi$ is to make a connection with the approximated distribution for $\pi$ introduced in Section 4.2. However, the authors state that $\tau \rightarrow 0$ so everything is deterministic. Why introduce $\pi$ while the approximated decision can be more straightforwardly expressed as
$$
{S_t}^* = \text{argmin} \sum_{i \in S_t} \text{margin}_t(i)?
$$

5. It is very easy for the authors to fail to understand the differences between Equations (8) and (10). In the former case, the perturbation set $S_t$ changes both the positions to be masked, and the positions to be summed over ($\sum_{i \in S_t}$). In the latter case, $I_t$ only changes the positions to be summed over, GIVEN the perturbation set fixed to $S_t$. This confusion is partly due to not emphasizing that the $\ell_i$ is a function of $S_t$. It would make the paper clearer if $\ell_i$ is changed to $\ell_i(S_t)$ in Equations (2), (3), and (10).

6. (minor) In line 273 (right), use \citet instead of \cite or \citep because Kim et al is used as a sentence component. So it should be 'following Kim et al. (2025b)'.

7. (minor) Font size in tables is too small. Consider changing it back to normal size, should there be additional space allowed.

---

> ### Author Rebuttal · Authors · 2026-03-31
>
> We thank the reviewer for the positive evaluation and attention to detail. We address the precise observations regarding notation as follows.
>
> > **Point 1:** The optimal solution to Equation 8 is concentrated on the discrete set $S_t$, making the continuous formulation $\pi$ potentially unnecessary.
>
> **Response to Point 1:**
> While the optimal solution ultimately requires a discrete mask, the continuous distribution $\pi$ explicitly models the uncertainty of perturbing each token. A continuous formulation allows the framework to assign masking probabilities to candidate positions proportional to their instability. Taking the limit $\tau \to 0$ then projects these probabilities into the deterministic mask $S_t$. The progression from a general probabilistic foundation to a specific discrete mask ensures the approximation is derived from a principled foundation rather than being an isolated heuristic. We have updated Sections 4.1 and 4.2 to strengthen this continuous-to-discrete bridge.
>
> > **Point 2:** The instability score $\ell_i$ depends on the chosen perturbation mask $S_t$, but the notation does not reflect this dependence.
>
> **Response to Point 2:**
> Thanks for pointing this out; we have updated the notation in Equations (2), (3), and (10) from $\ell_i$ to $\ell_i(S_t)$ to formalize the dependence of the instability metric on the specific chosen subset $S_t$.
>
> > **Point 3:** The conditioning notation $p_\theta(y_i^{(t)}=v | y^{(t)}, i)$ in the Preliminaries contains a mathematical circularity by conditioning on a state that already includes the position being predicted.
>
> **Response to Point 3:**
> Thanks for pointing this out; we have updated the notation to condition on $y_{\setminus i}^{(t)}$, denoting the state where position $i$ is masked. The update eliminates the mathematical circularity in the Preliminaries.
>
> > **Point 4:** Equation 11 omits the positional index required for the predicted token probability.
>
> **Response to Point 4:**
> Thanks for pointing this out; we have updated Equation 11 to $p_\theta(y_i^{(t)}=v | \tilde{y}_{\setminus i}^{(t)})$ to indicate positional index.
>
> ---
>
> Furthermore, we have increased the size of the tables and fixed the citation.

---

> > ### Author Rebuttal · Reviewer_HGzu · 2026-04-02
> >
> > Writing issues have been largely addressed. I am adjusting my score.

---

> > > ### Author Response · Authors · 2026-04-07
> > >
> > > We thank the reviewer for updating the evaluation score. We appreciate the time dedicated to reviewing the manuscript and the constructive feedback regarding the mathematical notation.

---

### Official Review · Reviewer_QUft · 2026-03-10

**Soundness:** 3
**Presentation:** 3
**Significance:** 3
**Originality:** 3
**Overall Recommendation:** 4
**Confidence:** 4

**Summary:**

CoRe addresses a real weakness in masked diffusion decoding: tokens committed early under sparse context become hard anchors that propagate structural errors. Rather than measuring uncertainty at sampling time, CoRe probes whether each token remains probable when nearby tokens are jointly masked—treating instability under this perturbation as the revision signal. The method is training-free, adds ~6% compute overhead, and shows consistent gains on LLaDA-8B-Base, most strikingly +9.2% on MBPP.

**Compliance With Llm Reviewing Policy:**

Affirmed.

**Key Questions For Authors:**

1> Does instability-based revision improve performance when candidates are selected randomly or by recency rather than by low-margin? If not, is the method better characterized as a two-stage hybrid where margin does the heavy lifting?

2> What fraction of tokens that CORE actually revises would have been missed by the low-margin candidate screen?

3> What evidence supports that the chosen perturbation (masking low-margin tokens) approximates the worst-case context shift more closely than an arbitrary feasible perturbation?

4> Does jointly re-evaluating newly unmasked positions after a revision step change performance, and if so by how much?

**Limitations:**

I think the paper should more explicitly address two limitations:

1> The paper claims to move beyond confidence-based heuristics, yet the candidate pool is itself assembled using the top-2 probability margin which is a confidence-adjacent signal. This tension is never acknowledged or discussed.

2> The evaluation uses a single model (LLaDA-8B-Base) under greedy decoding with fixed generation length. The paper does not flag this as a scope limitation, yet several claims imply generality that the current evidence does not support.

**Strengths And Weaknesses:**

Strength:

1> The brittle-vs-uncertain distinction is conceptually neat and maps to a real failure mode. The qualitative examples (Appendix E) make the mechanism concrete in a way tables alone cannot.

2> Compute-controlled comparisons (Tables 2–3) are handled carefully. That ReMDM-conf degrades on code when given equal compute is one of the paper's clearest results.

3> The code generation gains are large enough to matter in practice for anyone working on MDM deployment.

Weakness:

1> The algorithm identifies candidates by low-margin, then ranks by instability, but these stages are not separated. Table 2's controls all share the same margin-based candidate pool, leaving open whether instability or the margin prefilter drives the gains. This missing ablation simultaneously weakens soundness, originality, and the central framing.

2> The appendix result confirms that the computed instability is a lower bound on worst-case instability—the chosen perturbation is feasible, not near-optimal. Section 4.1 should be framed as motivation, not justification.

3> All results come from one model (LLaDA-8B-Base) under greedy decoding; claims of broader generality are not necessarily supported.

---

> ### Author Rebuttal · Authors · 2026-03-31
>
> We thank the reviewer for the positive evaluation. We address the pre-filter ablation, theoretical framing, and evaluation scope below.
>
> > **W1, KQ1, & Lim1: Performance of random or recency-based candidate selection**
>
> Prior methods use confidence scores to dictate revisions. In contrast, CORE utilizes the margin strictly to filter candidates, relying on probability mass under context perturbation to dictate the final revision. Applying instability scoring to arbitrary candidate pools isolates this effect:
>
> Candidate Selection|HumanEval|MBPP
> -|-|-
> Base Model (No Revision)|12.80%|15.60%
> Recent + CORE|12.80%|18.00%
> Random + CORE|14.63%|22.40%
> Low Margin + CORE|17.07%|24.80%
>
> Because diffusion models assign high probability mass to local syntax, random selection wastes compute on structurally stable positions. The framework instead uses the margin to filter specifically for tokens with distributed probability mass. Table 2 isolates the contribution of instability: selecting revisions via margin alone (Margin Remask) yields 17.40% MBPP, whereas using the margin as a screen for CORE reaches 24.80%. This 7.4% accuracy gap indicates instability provides the signal required for sequence recovery.
>
> > **W2 & KQ3: Evidence that masking low-margin tokens approximates worst-case context shifts**
>
> Section 4.1 frames the continuous formulation as motivation. To justify the margin screen, Theorem 1 establishes the relationship between top-2 margin and contextual sensitivity:
>
> **Theorem 1: Small top-2 margin implies high contextual sensitivity.**
> Let $i \in C_t$ be a candidate token position, and let $y_{\setminus i}^{(t)}$ denote the sequence with position $i$ masked. Let $v_i^1, v_i^2$ be the top-1 and top-2 tokens under the distribution $p_\theta(v \mid y_{\setminus i}^{(t)})$. Assume the conditional distribution is Lipschitz continuous with respect to context perturbations, bounded by a distance metric $d(\tilde y_{\setminus i}^{(t)}, y_{\setminus i}^{(t)})$ and a uniform Lipschitz constant $L$:
> $$|p_\theta(v \mid \tilde y_{\setminus i}^{(t)})-p_\theta(v \mid y_{\setminus i}^{(t)})|\le L \cdot d(\tilde y_{\setminus i}^{(t)}, y_{\setminus i}^{(t)}), \quad \forall v\in\mathcal V$$
> If $2L \cdot d(\tilde y_{\setminus i}^{(t)}, y_{\setminus i}^{(t)}) < \text{margin}_t(i)$, the top-1 token at position $i$ remains invariant under the perturbation.
>
> **Proof of Theorem 1:**
> Inverting the top-1 prediction requires $p_\theta(v_i^2 \mid \tilde y_{\setminus i}^{(t)}) \ge p_\theta(v_i^1 \mid \tilde y_{\setminus i}^{(t)})$. Under the Lipschitz constraint:
> $$p_\theta(v_i^1 \mid \tilde y_{\setminus i}^{(t)}) \ge p_\theta(v_i^1 \mid y_{\setminus i}^{(t)}) - L \cdot d$$
> $$p_\theta(v_i^2 \mid \tilde y_{\setminus i}^{(t)}) \le p_\theta(v_i^2 \mid y_{\setminus i}^{(t)})+L \cdot d$$
> where $d:=d(\tilde y_{\setminus i}^{(t)}, y_{\setminus i}^{(t)})$. Therefore:
> $$p_\theta(v_i^1\mid\tilde y_{\setminus i}^{(t)})-p_\theta(v_i^2 \mid\tilde y_{\setminus i}^{(t)})\ge (p_\theta(v_i^1 \mid y_{\setminus i}^{(t)})-p_\theta(v_i^2\mid y_{\setminus i}^{(t)})) - 2L \cdot d = \text{margin}_t(i) - 2L \cdot d$$
> If $2L \cdot d < \text{margin}_t(i)$, then $p_\theta(v_i^1 \mid \tilde y_{\setminus i}^{(t)})>p_\theta(v_i^2 \mid\tilde y_{\setminus i}^{(t)})$, preventing a top-1 token inversion.
>
> Theorem 1 establishes that the distance $d$ to alter maximal probability is bounded by the top-2 margin. High-margin positions theoretically resist inversion under small context shifts. Targeting low-margin tokens isolates structural vulnerabilities, providing a grounded approximation of the worst-case objective.
>
> > **KQ2: Fraction of tokens missed by the low-margin candidate screen**
>
> Because high-margin positions resist perturbation (Theorem 1), instability scores theoretically remain below the revision threshold, rendering the missed fraction negligible. Calculating a sequence-wide fraction is mathematically ill-posed due to **trajectory divergence**: revising a token at step $t$ alters the context for subsequent steps, invalidating direct 1-to-1 trajectory comparisons. Table 2 empirically indicates the candidate screen does not bottleneck quality (+7.4% accuracy over margin-only selection).
>
> > **KQ4: Performance impact of jointly re-evaluating newly unmasked positions**
>
> Jointly re-evaluating unmasked positions degrades accuracy and doubles cost per revision:
>
> Benchmark|CORE (No Re-eval)|With Re-eval
> -|-|-
> HumanEval|17.07%|16.46%
> MBPP|24.80%|24.00%
>
> Reusing initial forward-pass probabilities yields higher accuracy while preserving computational resources.
>
> > **W3 & Lim2: Generality beyond single model and greedy decoding**
>
> Dream-7B evaluations demonstrate cross-model generality:
>
> Benchmark|Dream-7B|Dream-7B + CORE
> -|-|-
> HumanEval|20.10|29.26 (+9.16)
> MBPP|25.60|34.60 (+9.00)
>
> Under stochastic decoding (Appendix Table 6; temperature=1.0), the mean accuracy on MBPP (25.24%) exceeds the baseline (18.40%) by 4 standard deviations.

---

> > ### Author Rebuttal · Reviewer_QUft · 2026-04-02
> >
> > I think the author has adequately addressed most issues (except limitation) I raised.

---

> > > ### Author Response · Authors · 2026-04-07
> > >
> > > We thank the reviewer for evaluating the rebuttal. To address the limitation request, the final manuscript will include a dedicated paragraph detailing the boundary between the margin-based candidate screen and the instability metric. We appreciate the constructive feedback.

---

### Official Review · Reviewer_DPxB · 2026-03-13

**Soundness:** 4
**Presentation:** 2
**Significance:** 3
**Originality:** 3
**Overall Recommendation:** 4
**Confidence:** 4

**Summary:**

This paper studies revision in diffusion language models. The authors argue that early tokens can be wrong even when they look confident. Their method tests whether a token stays reliable after part of the context is masked. If the token becomes unstable, the model revises it. The method helps most on code generation tasks.

**Compliance With Llm Reviewing Policy:**

Affirmed.

**Final Justification:**

I thank the authors for their detailed response. My concerns have been adequately addressed.

**Key Questions For Authors:**

see weakness above

**Limitations:**

yes

**Strengths And Weaknesses:**

### Strengths
1. Clear and well-motivated problem setup.
2. Training-free and easy to plug into decoding.
3. Strong gains on code benchmarks, especially MBPP.

### Weaknesses
1. The practical method still relies on heuristic margin-based candidate selection.
2. The robust optimization story is only approximately realized in the final algorithm.
3. Validation is limited mainly to one base model, LLaDA-8B-Base.

---

> ### Author Rebuttal · Authors · 2026-03-31
>
> We thank the reviewer for recognizing the clear motivation, training-free nature, and strong empirical gains of our framework. We address the specific points below.
>
> > **W1:** The practical method still relies on heuristic margin-based candidate selection.
>
> **Response to W1:**
>
> To isolate the instability metric from the margin heuristic, Table 2 of the main paper compares the Margin Remask baseline against CORE. Relying solely on the margin heuristic to select revisions yields a 17.40% accuracy on MBPP. Using CORE's instability score to choose which tokens to revise increases accuracy to 24.80%.
>
> To demonstrate that the instability metric drives performance across arbitrary candidate distributions, we evaluated alternative proposal strategies.
>
> Candidate Selection | HumanEval | MBPP
> --- | --- | ---
> Base Model (No Revision) | 12.80% | 15.60%
> Recent + CORE | 12.80% | 18.00%
> Random + CORE | 14.63% | 22.40%
> Confidence + CORE | 15.24% | 25.00%
> Low Margin + CORE | 17.07% | 24.80%
>
> CORE improves performance across all candidate selection strategies. Applying the active context perturbation to a random subset of tokens yields a +6.80% absolute improvement over the base model on MBPP (from 15.60% to 22.40%). This shows that the instability metric, rather than the heuristic proposal mechanism, is primarily responsible for improvements. The low-margin criterion yields the highest cross-benchmark performance, justifying its use in the primary configuration.
>
> > **W2:** The robust optimization story is only approximately realized in the final algorithm.
>
> **Response to W2:**
> Identifying the exact worst-case perturbation over combinatorial subsets in Distributionally Robust Optimization (DRO) is computationally intractable, necessitating the transition from the continuous formulation to the discrete algorithm. Bypassing the combinatorial search via continuous relaxations mirrors established optimization methodologies in discrete adversarial robustness (Zou et al., 2023; Ebrahimi et al., 2018).
>
> While the final algorithm is a discrete approximation, the DRO formulation is required to establish the bounds of the algorithm. As demonstrated in Appendix C, the continuous theoretical formulation provides the structure necessary to show that evaluating the instability metric on the deterministic subset $S_t$ yields a certified lower bound on the exact worst-case instability. Therefore, the theoretical framework serves as the proof that the discrete algorithm optimizes a certified lower bound of the intractable objective.
>
> We added a paragraph to Section 4.1 detailing the proof connecting the continuous formulation to the deterministic approximation.
>
> [1] Zou, Andy, et al. "Universal and transferable adversarial attacks on aligned language models." arXiv preprint arXiv:2307.15043 (2023).
> [2] Ebrahimi, Javid, et al. "Hotflip: White-box adversarial examples for text classification." Proceedings of the 56th Annual Meeting of the Association for Computational Linguistics. 2018.
>
> > **W3:** Validation is limited mainly to one base model, LLaDA-8B-Base.
>
> **Response to W3:**
> We now have additionally evaluated CORE on the Dream-7B base model.
>
> Method | HumanEval | MBPP
> -|---|-
> Dream-7B | 20.10 | 25.60
> Dream-7B + CORE | 29.26 (+9.16) | 34.60 (+9.00)
>
> CORE's revision on the Dream-7B architecture improves HumanEval by +9.16 points and MBPP by +9.00 points. We have added this to the experiments.

---

> > ### Author Rebuttal · Reviewer_DPxB · 2026-04-01
> >
> > I thank the authors for their detailed response. My concerns have been adequately addressed.
> >
> > I believe my current score remains appropriate.
> >
> > Good luck!

---

> > > ### Author Response · Authors · 2026-04-07
> > >
> > > We thank the reviewer for confirming the rebuttal addresses the initial concerns. We appreciate the time dedicated to evaluating the manuscript.

---

### Official Review · Reviewer_pPkG · 2026-03-16

**Soundness:** 3
**Presentation:** 2
**Significance:** 2
**Originality:** 3
**Overall Recommendation:** 3
**Confidence:** 4

**Summary:**

This paper proposes a remasking procedure for discrete diffusion models that remasks tokens that are sensitive to other tokens being remasked. The procedure choses a set of remasking candidates using the margin between the top-1 and top-2 probabilities, then computes an instability score over those candidates. The instability score is simply the surprisal of the unmasked token under the perturbed context. Experiments show some gains primarily on coding tasks.

**Compliance With Llm Reviewing Policy:**

Affirmed.

**Key Questions For Authors:**

Can you explain what Figure 3 is showing? I interpret it as the instability score cleanly separates tokens. But it does not provide much evidence on whether instability is actually the right measure. Ideally the analysis section would contain evidence that instability is the right measure.

I can imagine a setting where instability is not the right measure to resample: sudoku with a diffusion model. Difficult tokens might be unstable initially, but as search proceeds they would become stable. Back-tracking would require re-masking a large number of tokens. While this is a common weakness of masked diffusion with remasking, it still shows the shortcomings of instability as a metric.

**Limitations:**

Yes

**Strengths And Weaknesses:**

The core idea makes intuitive sense as an informal heuristic. Informally, I would want a procedure that finds tokens with high mutual information with only a small number of existing tokens, and retries those together. This should be an improvement over previous remasking procedures, which do not incorporate second-order information. This paper pushes towards that.

Soundness and presentation: The paper appears correct and is well-written for the most part. There is some over-formalization. Some notation in the main paper (Q in equation 6) is barely used and not included in the notation table 5. This could easily be replaced with a single phrase, such as "a distribution over perturbed contexts". Additionally, while the improvements are significant over the baseline, the baseline's code generation numbers are extremely low compared to modern LLMs which have the benefit of on-policy distillation from stronger models.

Significance: This is a significant problem for masked diffusion models, although arguably I expected remasking to ultimately be learned through reinforcement learning rather than step-wise heuristics. This paper seems to make modest progress on this front. The biggest weakness is the experimental results. A secondary weakness is the lack of justification for instability as the target measure.

Originality: I believe their approach is novel.

---

> ### Author Rebuttal · Authors · 2026-03-31
>
> We thank the reviewer for the thoughtful evaluation and for recognizing that the framework provides a novel, intuitive improvement over prior remasking procedures by incorporating second-order information. We address the concerns regarding the metric justification, presentation, and evaluation scope below.
>
> > **W1:** Over-formalization. Some notation in the main paper ($Q$ in equation 6) is barely used and not included in the notation table.
>
> **Response to W1:**
> We thank the reviewer for highlighting the need for better intuitive grounding. Following your feedback, we have now removed the $Q$ notation from Equation 6, Equation 7, and the surrounding text. We replaced it with a simplified expectation taken directly over the binary mask distribution $p_\pi(z)$, streamlining the mathematical presentation without altering the robust optimization objective.
>
> > **W2:** While the improvements are significant over the baseline, the baseline's code generation numbers are extremely low compared to modern LLMs which have the benefit of on-policy distillation from stronger models.
>
> **Response to W2:**
> Our paper addresses revision in popular diffusion language models such as LLaDA rather than the autoregressive setting. In addition, we highlight CORE is not tied to the LLaDA architecture by evaluating CORE on the **Dream-7B** base model, observing similar performance gains of approximately +9 points on both HumanEval and MBPP.
>
> |                 | HumanEval     | MBPP |
> |----------       |----------     |------|
> | Dream-7B        | 20.10         | 25.60|
> | Dream-7B + CORE | 29.26 (+9.16)| 34.60 (+9.00)|
>
> > **W3 & Q1:** Can you explain what Figure 3 is showing? I interpret it as the instability score cleanly separates tokens. But it does not provide much evidence on whether instability is actually the right measure.
>
> **Response to W3 & Q1:**
> Your understanding of Figure 3 is correct; the empirical distribution depicts how structurally stable tokens group near zero, while context-brittle tokens exhibit heavy-tailed instability.
>
> In regards to instability as the right measure, concurrent work on self-correction in masked diffusions (PRISM; Kim et al., 2025) shows that an optimal token quality score is proportional to the token's marginal probability under context perturbation. CORE's instability score computes this objective at inference time, providing a theoretically grounded, training-free alternative to learned RL verifiers and scorers.
>
> We justify the metric empirically via the **Margin Remask** baseline in Table 2. Compared to CORE, Margin Remask revises tokens with smallest margin rather than largest instability. As detailed in Section 5.2, a token can exhibit a low margin because multiple lexical alternatives are acceptable (e.g., swapping "result" for "output"). Revising the low-margin tokens without stability assessment yields a +1.8 accuracy improvement on MBPP. In contrast, CORE ranks the low-margin candidates by likelihood under the jointly-masked perturbed context. This essentially means: ranking by likelihood targets conditional dependence, identifying tokens tied to hard consistency constraints (e.g., variable bindings) rather than isolated vocabulary choices. The CORE mechanism yields a +9.2 points improvement. The performance difference isolates probability mass instability as the signal required to resolve structural errors.
>
> [1] Kim, Jaeyeon, et al. "Fine-Tuning Masked Diffusion for Provable Self-Correction." arXiv preprint arXiv:2510.01384. 2025.
>
> > **Q2:** I can imagine a setting where instability is not the right measure to resample: sudoku with a diffusion model. Difficult tokens might be unstable initially, but as search proceeds they would become stable.
>
> **Response to Q2:**
> This is an interesting point regarding Sudoku. We have evaluated CORE on the Sudoku benchmark from d1 (Zhao et al., 2025) as follows.
>
> | Benchmark | Base LLaDA | CORE |
> | --- | --- | --- |
> | Sudoku | 8.0% | 8.0% |
>
> Both the base model and CORE achieve 8.0% accuracy. This highlights that CORE does not force revisions when unnecessary and maintains baseline performance on Sudoku while delivering a +9.20% absolute gain on open-ended code generation (MBPP). We added a limitations paragraph to the appendix detailing this.
>
> [2] Zhao, S., Gupta, D., Zheng, Q., & Grover, A. (2025). d1: Scaling Reasoning in Diffusion Large Language Models via Reinforcement Learning. arXiv [Cs.CL]. Retrieved from http://arxiv.org/abs/2504.12216

---

### Decision · Program_Chairs · 2026-04-30

**Decision:**

Accept (regular)

**Comment:**

CoRe proposes a training-free remasking framework for masked diffusion models that identifies context-brittle tokens via adversarial perturbation rather than static confidence scores, with strong gains on code generation. Three reviewers converged to weak accept or accept with all concerns fully resolved; Reviewer pPkG remains at weak reject but did not engage during the discussion phase, and the rebuttal empirically addressed their metric justification and Sudoku edge-case concerns.